# Acceptability of daily pre-exposure prophylaxis among adolescent men who have sex with men, *travestis* and transgender women in Brazil: A qualitative study

Eliana Miura Zucchi[1]☯*, Marcia Thereza Couto[2]☯, Marcelo Castellanos[3]☯, Érica Dumont-Pena[4‡], Dulce Ferraz[5‡], Thiago Félix Pinheiro[2], Alexandre Grangeiro[2‡], Luís Augusto Vasconcelos da Silva[6], Inês Dourado[3], Leo Pedrana[3], Fernanda Soares de Resende Santos[7], Laio Magno[3,8]☯

1 Programa de Pós-Graduação em Saúde Coletiva, Mestrado Profissional em Psicologia e Políticas Públicas, Universidade Católica de Santos, São Paulo, Santos, Brazil, 2 Faculdade de Medicina, Universidade de São Paulo, São Paulo, São Paulo, Brazil, 3 Instituto de Saúde Coletiva, Universidade Federal da Bahia, Salvador, Bahia, Brazil, 4 Departamento de Enfermagem Materno Infantil e Saúde Pública, Universidade Federal de Minas Gerais, Belo Horizonte, Minas Gerais, Brazil, 5 Escola FIOCRUZ de Governo, Fundação Oswaldo Cruz, Brasília, Distrito Federal, Brazil, 6 Instituto de Humanidades, Artes e Ciências, Universidade Federal da Bahia, Salvador, Bahia, Brazil, 7 Programa de Pós-Graduação em Enfermagem, Universidade Federal de Minas Gerais, Belo Horizonte, Minas Gerais, Brazil, 8 Departamento de Ciências da Vida, Universidade do Estado da Bahia, Salvador, Bahia, Brazil

☯ These authors contributed equally to this work.
‡ These authors also contributed equally to this work.
* eliana.zucchi@uol.com.br

**Data Availability Statement:** Data cannot be shared publicly because of their sensitive content.

## Abstract

### Background

Adolescents face socio-structural, personal and programmatic barriers to HIV prevention services, highlighting the importance of understanding knowledge and acceptability as essential aspects to promote their broader access to pre-exposure prophylaxis (PrEP). We analyzed the acceptability of PrEP among adolescent men who have sex with men (MSM), *travestis* and transgender women (TGW).

### Methods

A qualitative investigation was conducted as part of the formative research of the PrEP15-19 study, an ongoing demonstration study that analyzes the effectiveness of daily PrEP among adolescent MSM, *travestis* and TGW aged 15–19 in three Brazilian cities. A total of 37 semi-structured interviews and 6 focus groups were conducted. Building from thematic analysis focusing on participants' sexual encounters, perceptions about PrEP efficacy, and vulnerability contexts, we analyzed prospective acceptability of PrEP.

### Findings

Knowledge about PrEP was incipient and characterized by adolescents' frequent doubts about its prescription and efficacy. The 'ideal' use of PrEP appeared together with consistent

The informed consent process prior to participation ensured confidentiality and anonymity and that only investigators of PrEP 1519 study would be allowed to access the data collected. With these conditions assured, ethical approval was obtained from the Research Ethics Review Committees of the Universidade de São Paulo (protocol number 70798017.3.0000.0065), Universidade Federal da Bahia (protocol number 01691718.1.0000.5030) and Universidade Federal de Minas Gerais (protocol number 17750313.0.0000.5149). In the best interest of protecting participants' confidentiality and anonymity, researchers may contact the Research Ethics Committee of Universidade de São Paulo, (Comissão de Ética para Análise de Projetos de Pesquisa, email: cappesq.adm@hc.fm.usp.br), to make requests related to access to the data used for the analyses in this manuscript.

**Funding:** ID and AG are principal investigators of the PrEP 1519 Study ('A Demonstration Project of the effectiveness HIV Pre-exposure Prophylaxis (PrEP) amongst adolescent men who have sex with men, transgender women at high risk for HIV infection in the context of Combination Prevention in Brazil') in the cities of Salvador and São Paulo, respectively, which is funded by Unitaid (grant number 2017-15-FIOTECPrEP). PrEP 1519 Study is also funded by the Brazilian Ministry of Health, through the Department of Chronic Diseases and Sexually Transmitted Infections, Bahia State Department of Health, São Paulo State and City Department of Health, and City of São Paulo AIDS Program, by donating PrEP medications, condoms, and rapid tests and providing the necessary infrastructure for the study development.

**Competing interests:** The authors have declared that no competing interests exist.

condom use, especially in casual sex. PrEP use was also mentioned as depending on increased learning about prevention management over time. Main barriers to PrEP use included the incorporation of a daily medication into participants' routine and its impact on their social lives, especially related to stigma. Concerns over short- and long-term side effects were also reported as barriers to PrEP use. TGW and *travestis* contrasted using PrEP with the precarity of their life conditions, and some expressed a critical vision about PrEP by associating it with pharmaceuticalization and trans necropolitics.

## Conclusions

Participants' low knowledge and acceptability of PrEP are circumscribed by a rigid perception of condom as the ideal prevention method and the context of their sexual relations. Prospective acceptability highlights that the successful uptake of PrEP depends on overcoming barriers of access to health services and confronting transphobia and homophobia as part of care.

## Introduction

In many countries including Brazil, adolescent men who have sex with men (MSM) and transgender women (TGW) are disproportionally affected by HIV infection. The literature describes different personal, social and structural barriers to adolescents' access to HIV services including prevention services and delays in diagnosis and treatment [1]. Personal barriers include the fear of being discriminated, the lack of access to quality information and the absence of social and family support in taking care of sexual health [2–4]. Power structures that legitimate social exclusion and violence against MSM and TGW produce a series of stigmatizing and discriminatory processes that result in critical socio and structural barriers to HIV prevention and care [4–6]. When such structural processes take plane in health delivery environments, examples of programmatic barriers faced by these populations include limited health clinic hours, discriminatory attitudes by health professionals and flaws in the confidentiality of health services [7–10].

Knowledge about pre-exposure prophylaxis (PrEP) varies substantially according to the context. In low and middle-income countries, the prevalence of knowledge about PrEP among MSM is 29.7% [11]. In Rio de Janeiro, Brazil, this percentage is 38.7% [12]. Although in the United States knowledge of PrEP among MSM recruited in demonstration studies is estimated to be 58.6% [13], this rate dropped to 28% in a community highly affected by HIV in New York City. Although knowledge among MSM and TGW can still be considered low, it has been increasing as PrEP programs have been expanded in some countries [14–17]. Nonetheless, important knowledge gaps about PrEP remain in key populations [16, 18, 19]. Factors that contribute to higher PrEP knowledge and awareness among MSM include high levels of schooling and living in large cities [19], belonging to with LGBT networks [20, 21], and prior HIV testing [12].

On the other hand, high acceptability of PrEP has been observed in many studies, and main facilitators were free access to PrEP [2, 17, 20], as well as access to prevention services that provide free HIV testing and sexual health follow-up on a regular basis, and individual counseling [2]. The Brazilian Health System (in Portuguese *Sistema Único de Saúde–SUS*) offers free and universal access, which includes HIV prevention, treatment, and care. Daily oral PrEP was

introduced to public health services in late 2017 for populations (individuals aged 18 or above) at substantial risk for HIV infection: gay and other MSM, transgender people, sex workers and serodiscordant couples [22].

Knowledge and acceptability are essential aspects of promoting broader access to PrEP among adolescents. Other relevant issues for PrEP acceptability among this population are personal autonomy, sexual-affective relationships, disclosure of sexual orientation, gender identity and bodily changes produced by biomedical technologies. In addition, adolescents may be more severely affected by stigma, discrimination, violence, exclusion and poor health conditions [23].

Understanding how adolescents deal with the information about PrEP, its potential acceptability and suitability, as well as how their sexual encounters intersect with their life contexts, can permit to identify the intention of use and potential obstacles to access to this technology. These data can also produce information to support the planning of strategic actions to expand access, linkage to health services and adherence to PrEP among these populations. In this paper, our goal was to analyze the acceptability of daily oral PrEP among adolescent MSM, *travestis* and TGW from three Brazilian capital cities.

## Methods

### Participants

Data were obtained as part of the formative research (FR) of the PrEP1519 an ongoing demonstration study on the effectiveness of daily PrEP among adolescent MSM, *travestis* and TGW aged 15 to 19 years old. PrEP enrolment initiated in March 2019 in the city of São Paulo, and in April in Salvador and Belo Horizonte. Before participants' enrolment, we conducted a FR to investigate dynamics of social interaction, sexual experiences of sociability, and acceptability of HIV prevention methods and recruitment strategies. We adopted a FR framework similar to others used in respondent-driven sampling surveys among hard-to-reach populations using mapping out of social spaces, individual interviews and focus group discussions (FGD) [24]. As a first step, in each city, we mapped out adolescent venues with a high concentration of MSM, *travestis* and TGW, both in face-to-face settings (e.g., beaches, parks, bars, parties, sports and leisure facilities, schools, health and human rights services) and in online spaces (e.g., social media, internet, and dating apps). Participant observation was conducted in order to understand the youth venues dynamics and to identify potential adolescent key-informants. As a second step, between August 2018 and January 2019, we selected–mainly at bars, parties, schools and dating apps–key-informants to participate in semi-structured interviews and in FGD in the three aforementioned cities. Differently from other contexts in which the word 'transvestite' has a derogatory meaning, in Brazil, the term *travesti* represents a political identity, recognized as so by the LGBTQIA+ social movement. In respect to this political claim and to the self-identification of the study participants, in this article we used the terms *travesti* and transgender women, and, when we referred to both identities, we used the term 'transgender and gender diverse' (TGD).

### Materials and procedures

A total of 58 adolescents participated in the FR, of whom 26 adolescents participated both in interview and FGD.

Thirty-seven adolescent MSM, *travestis* and TGW accepted to be interviewed, including 16 in Salvador, 13 in São Paulo and 8 in Belo Horizonte. Semi-structured interviews covered the following topics: (a) networks and social venues, (b) gender identity/expression, sexual orientation, and sexuality, (c) vulnerability to HIV, violence, and discrimination, (d) knowledge of

and motivation to use PrEP and self-testing and (e) perceptions on the recruitment strategies of the PrEP15-19 study. The interview guide can be found as S1 File.

Two FGD were conducted in each city, of which one was with TGD and the other with MSM. The composition was: 5 TGD and 10 MSM in Salvador, 03 TGD and 09 MSM in São Paulo and 4 TGD and 06 MSM in Belo Horizonte. FGD started with a small explanation about PrEP and the PrEP15-19 study design and recruitment strategies and were followed by questions to encourage discussion about these topics. The guide for the FGD can be found as S2 File.

During the interviews and FGD, PrEP was explained as: "Pre-Exposure Prophylaxis is an HIV prevention strategy. A person who doesn't have HIV takes antiretroviral drugs (the same pills people take to treat HIV) because they have unprotected sex (like sex without a condom). If the pills are taken every day, it will prevent HIV infection. If the person does not take it every day, they may get infected with the virus."

Interviews and FGD were conducted in Portuguese, in private rooms, by trained researchers, and lasted on average 90 and 120 minutes, respectively.

## Data analysis

Interviews were audio-recorded and transcribed. A researcher from the coordinating team at each site assessed the accuracy of the transcripts by reviewing selected interviews and comparing the audio and the transcripts. Quotes presented in this paper were translated into English by a professional translator and were reviewed by the authors. Thematic analysis [25] covered the following stages: the reading of the transcripts in detail; definition of thematic categories thought the discussion between the research team of the three sites which was based on the more frequent topics of the narratives as well as from the scientific literature on the subject; the extraction of the content of the interviews according to the categories was done by two researchers in each study site; the analysis of each category in dialogue with the current literature and theoretical references on knowledge and acceptability of PrEP. For purposes of anonymity, participants' names were changed using code numbers and their quotes are characterized by target population (MSM or TGD), participant number, age, study site (SP, SSA, and BH), and technique (interview or FGD).

We analyzed the prospective acceptability of PrEP using the Theoretical Framework of Acceptability (TFA) [26]. The acceptability analysis included data about the participants' sexual encounters, their perceptions about the efficacy of PrEP and their contexts of vulnerability for HIV infection. The analysis of acceptability was prospective since PrEP had been made available at Brazilian public health system in late 2017 to individuals aged 18 or above, and none of the participants had ever used it.

Based on these central themes, we developed thematic categories that we then related to the TFA. The thematic categories were: limited knowledge of PrEP and how it works; perception of PrEP efficacy; PrEP as a counterpoint to condoms in narratives about prevention; aspects of age difference; barriers to PrEP use; pharmaceuticalization and trans necropolitics in the relationship between the State and the Market; and stigmas related to HIV, adolescence and sexual orientation.

Following the TFA, each thematic subcategory was analyzed in terms of seven component constructs of acceptability: 1. Affective attitude, defined as "how an individual feels about taking part in an intervention"; 2. Burden, defined as "the perceived amount of effort that is required; 3. Opportunity costs, defined as "the extent to which benefits, profits, or values must be given up to engage in an intervention"; 4. Ethicality, defined as "the extent to which the intervention has good fit with an individual's value system"; 5. Self-efficacy, defined as "the

participant's confidence that they can perform the behavior(s) required to participate in the intervention"; 6. Intervention coherence, defined as "the extent to which the participant understands the intervention, and how the intervention works"; and 7. Perceived effectiveness, defined as "the extent to which the intervention is perceived as likely to achieve its purpose" [26]. As we propose the acceptability construct to indicate intention to use PrEP, we analyzed narratives that could be interpreted as prospective acceptability of PrEP.

### Ethical statement

The formative research protocol was approved by the Research Ethics Review Committees of the Universidade de São Paulo (protocol number 70798017.3.0000.0065), Universidade Federal da Bahia (protocol number 01691718.1.0000.5030) and Universidade Federal de Minas Gerais (protocol number 17750313.0.0000.5149). All participants were informed of the research aims and their rights related to research participation. Individuals aged 18 y.o. or above signed an informed consent form. Individuals aged 15–17 signed an informed assent form. Based on the principle of non-maleficence, Research Ethics Review Committees waived parental consent as it might threaten adolescents' confidentiality regarding sexual orientation and/or gender identity thereby creating a risk of discrimination or violence.

All participants were invited to enroll in the PrEP1519 cohort study, through which they would have ample access to PrEP and other HIV prevention methods, such as condoms, testing, and post-exposure prophylaxis (PEP). Two participants tested positive for HIV during the FR and were referred to treatment services.

## Results

### The participants

Fifty-eight adolescents participated in the study, 40% were between 16 to 17 years old. As for gender identity, 53% self-defined as men, 22% as transgender women and 12% as *travesti*. Nearly half (52%) defined their sexual orientation as homosexual or gay (all men), 17% bisexual, 12% pansexual, and 12% heterosexual (all TGW and *travestis*). Most participants reported incomplete (47%) or complete (38%) high school. As for skin color, 55% self-identified as black or brown-mixed and, 38%, as white. Forty-five per cent referred prior HIV testing (Table 1).

Hiding or not talking about their sexual orientation was a recurring theme in the narratives of MSM. Narratives of others welcoming and accepting disclosures were an exception. TGD reported that gender-affirming processes had an immediate impact in distancing themselves from their families and school dropout due to transphobia.

The main contexts of sociability for friendship and finding partners included parties, funk dance parties, nightclubs, public venues and friends' house. Nearly half referred to use apps and social networks to find partners, regardless of their gender identity or sexual orientation.

### Limited knowledge of PrEP and how it works

Most (59%) participants had no previous knowledge about PrEP (Table 1). Those who had already heard about PrEP, obtained the information through internet, including online social networks and dating apps, social circles (LGBT or not), from people who used the prophylaxis, from health services or because of their direct participation in prevention programs.

**Table 1. Characterization of participants and their previous knowledge of PrEP.** São Paulo, Belo Horizonte, Salvador. Brazil. 2019.

| Characteristic | n | % |
|---|---|---|
| **Study site** | | |
| Belo Horizonte (BH) | 17 | 29.3 |
| Salvador (SSA) | 16 | 27.6 |
| São Paulo (SP) | 25 | 43.1 |
| **Age (years)** | | |
| 16–17 | 23 | 39.7 |
| 18–20 | 35 | 60.3 |
| **Gender identity** | | |
| Men | 31 | 53.4 |
| Transgender women | 13 | 22.4 |
| *Travesti* | 7 | 12.1 |
| Did not respond | 7 | 12.1 |
| **Sexual orientation** | | |
| Homosexual or gay | 30 | 51.7 |
| Pansexual | 7 | 12.1 |
| Heterosexual | 7 | 12.1 |
| Bisexual | 10 | 17.2 |
| Asexual | 1 | 1.7 |
| Did not respond | 3 | 5.2 |
| **Skin color/ ethnicity** | | |
| Black | 18 | 31.0 |
| White | 22 | 37.9 |
| *Parda* (brown/mixed) | 14 | 24.1 |
| Indigenous | 1 | 1.7 |
| Did not respond | 3 | 5.2 |
| **Schooling** | | |
| Incomplete high school | 27 | 46.6 |
| Complete high school | 22 | 37.9 |
| Incomplete undergraduate | 5 | 8.6 |
| Did not respond | 4 | 6.9 |
| **Has been tested for HIV** | | |
| Yes | 26 | 44.8 |
| No | 13 | 22.4 |
| Did not respond | 18 | 31.0 |
| Did not know | 1 | 1.7 |
| **Previous knowledge of PrEP** | | |
| No | 34 | 58.6 |
| Yes | 18 | 31.0 |
| Did not respond | 6 | 10.3 |

*I first had contact with PrEP on Grindr, there was a guy who was saying this and that about PrEP, and then I looked up information and I said, "guys, take a look at this". (MSM_07, 19 y.o., SSA, interview)*

Knowledge about PrEP was characterized by misinformation and frequent doubts about its prescription and efficacy, the need to use condoms and its interaction with alcohol, drugs and,

in the case of TGD participants, with hormones. There was also confusion between PrEP and post-exposure prophylaxis (PEP).

*There's counterindication so, for example, when you take PrEP, you can't use drugs. (MSM, SSA, FGD)*

*I'm starting to use hormones, I have to take three pills a day. So I was wondering if PrEP cuts the effect of hormones, what its side effects are. (TGW, SSA, FGD)*

*PrEP is a pill that should be taken after sex, I think that even with prevention [referring to condom use] it would be interesting to take it. (MSM_04, 17 y.o., SSA, interview)*

## Perception of PrEP efficacy

Many participants questioned whether PrEP actually protected from HIV infection as much as condom use did.

*Because I think that it's 50% safe. You have to use it, but you have to use condoms too. (TGW_03, 18 y.o., SSA, interview)*

*I don't know if it even eliminates the possibility of you being contaminated by HIV. But I know that it at least keeps it away. (MSM_02, 19 y.o., SSA, interview)*

On the other hand, it is noteworthy that there were a small number of participants that credited PrEP with additional protection against other STIs, thus showing a misconception related to PrEP indication.

*Imagine the benefit of a medication (referring to PrEP) that people from 15 to 19 years old could take to increase, for example, their immune system, (. . .) their immune system is going to be so strong that they'll practically be immune to a STD or even other types of more serious illnesses, imagine the benefit! So, this is basically why I would use PrEP. (MSM_08, 19 y.o., SSA, interview)*

## PrEP as a counterpoint to condoms in narratives about prevention

Prevention for adolescents was symbolized as using condoms in each and every sexual relation, especially in the context of casual sex, which was perceived as posing higher risk for STIs. The 'ideal' use of PrEP appeared together with condom use, either because condoms were seen as being indispensable by themselves or because participants considered the possibility of getting infected with other STIs.

*Condoms, even if you are taking PrEP, are indispensable. . . I think that it's difficult for people to understand this, and the use of PrEP ends up making this understanding difficult: 'I still need to use condoms' (. . .) because sometimes you feel safe, like you're immune to HIV, but you end up putting yourself at risk for other diseases, you know? (MSM, SP, FGD)*

*(Referring to the need of keeping using condoms while using PrEP) You've got to get the guy to put on a condom, we aren't obligated to get the diseases that he might have, and I say it just like that, I tell him to his face, "Handsome, you don't have a condom, keep on walking. . ." (TGD, BH, FGD)*

The rigidness of condom use as the 'ideal' prevention method faded when faced with the contingencies of the sexual-affective adolescent encounters. The perceived effectiveness of the

prophylaxis was mediated by affective attitudes towards not using condoms as much as they should, allowing for 'reckless behavior' and the safety and pleasure that PrEP could offer. Thus, PrEP appeared as an alternative in situations in which adolescents perceived limited self-efficacy for condom use.

This was also the case in transactional or paid sex, a recurrent experience in the narratives of TGW and *travestis*. In their narratives, PrEP was perceived as being potentially effective in situations of sexual violence in which clients demanded condomless sex, when the condom broke, and also to earn more money for sex without condoms. Participants also referred to cocaine use with clients and PrEP, which was seen as contributing to individual protection in such situations when they might be under the effect of something that alters levels of consciousness.

> *(Referring to the context of commercial sex) For these girls, that are on the street . . . that work as sex workers, right? I believe that it is also an evolution, because of course no one is going to remember. . . in these experiences, no one is going to remember to always use a condom as a lot of propositions and other things come up, right? So the pill will help a lot. (TGD, SP, FGD)*

In the context of casual sexual interactions that involved the use of alcohol and drugs, PrEP was referred to as a prevention possibility in case condoms were forgotten. As such, the perception of limited self-efficacy for condoms appeared to favor the acceptability of PrEP, which was seen as contributing to individual protection in such situations when they might be under the effect of something that alters levels of consciousness.

> *You're going to be drunk, right?! Your head is going to be mentally elevated, let's put it this way. . . you are going to be in the heat of the moment, making out and such. . . you are not going to worry about this. Not the same way as when you're in your sane state, so I think that it could influence a lot of things. (TGW_01, 16 y.o., BH, interview)*

In addition to these aspects, participants also mentioned the security of using PrEP when having sex with someone who was HIV-positive.

> *I know people that have, that got HIV without knowing that they were catching it because the condom broke and it ended up happening. With this medicine I think that that it is easier, even if the condom breaks or something happens, I would feel more protected. (TGD, BH, FGD)*

As such, participants perceived PrEP could provide additional security in specific situations where the effectiveness of condom use (self-efficacy) was uncertain, such as transactional sex, sex in the context of drug or alcohol use, casual sex and sex with people living with HIV. This feeling of security was referred to even considering side effects.

> *Yes, I thought that it would be. . . that would help a lot, because I would be more relieved. Not that I was going to have sex with other people without a condom, but I would be relieved, in case something unpredictable happened in the moment. (MSM_01, 20 y.o., SSA, interview)*

Although rare, the perception of PrEP as a way to abandon condom use also appeared in the narratives.

*I wouldn't even need to use condoms anymore because I would be using the medication. (MSM_06, 18 y.o., BH, interview)*

If, on one hand, PrEP was perceived of as potentially effective when condoms were not viable or failed, its ethicality was questioned by participants who saw the prophylaxis as an "easy" alternative for people who only thought about pleasure or that had sex "a lot". They also emphasized the possibility of developing addiction, by comparing PrEP to the use of medication for chronic diseases, thus arguing that PrEP should only be used as a "last resort."

*(referring to using PrEP) It's as if you were a diabetic, you can't be without insulin. . . almost like that (. . .) I think that yes, it causes addiction, but it has to do with how people see this dependence, if it is going to be good for their life or not. (TGD, SP, FGD)*

*The little fags must be thinking like this: 'Oh, I'm going to take this medication to be able to have tons of sex without condoms', because people who like penetration don't like penetration with condoms.' (MSM_02, 19 y.o., SSA, interview)*

### Aspects of age difference–increased learning about prevention management over time and interaction with adults when informed about PrEP

Some narratives reflected a perception of learning throughout sexual trajectories during adolescence and youth that influenced how prevention was planned.

MSM participants from one FGD illustrated this aspect when they regarded themselves as being too young to negotiate condom use and manage prevention when compared to men aged 20 and 21, who could be motivated to use PrEP due to their more developed capacity for planning and managing prevention. As such, self-efficacy was perceived to increase with learning about sexual life.

*Well yes, it depends, I think that if it was someone really young, like 18. . .(. . .) but if it was a more experienced person. . . they wouldn't need to be much older. . . 20, 21 or that had a more sexually active life. . . They would think about PrEP. (MSM, SP, FGD)*

Intergenerational differences among MSM adolescents and their parents about how they perceived prevention methods and recognized their sons were sexually active were referred to as challenges in conversations about prevention. Participants mentioned that their parents, possibly because they were only familiar with condoms, would not accept PrEP as a prevention option for their sons, which makes intervention coherence to be perceived differently between adolescents and their family.

*We have to think that people are going to have a different reaction. If I tell my mom that I am taking PrEP, because of HIV, she is simply going to say, "so, why are you having sex?" "You aren't using condoms?" (MSM, BH, FGD)*

### Barriers to PrEP use

Participants identified the incorporation a daily medication into their routine and its impact on their social lives as possible obstacles to PrEP use. Some also referred to PrEP as a "strong medicine" with negative side effects that could affect their bodies. For some, this compromised the acceptability of the prophylaxis. Some of those with prior knowledge about PrEP expressed worry and fear of its side effects in the present and future. Side effects mentioned included dry

mouth, difficulty with concentration and/or focus and possible negative effects on bones and kidneys.

Among some participants, their fears associated with the side effects of PrEP extrapolated physical and psychological dimensions into social arenas, such as their daily life and personal projects like their studies. TGD adolescents, in particular, mentioned the precarity of their life conditions, with uncertainty with regards to food and housing, as elements of their daily lives that contributed to their reticence towards PrEP as a viable option for them.

> *And then there's the entire context as well, because if a trans girl is taking the medicine, you don't know if the girl is eating everyday, how she's eating, if she's drinking water, if she's going to be able to rest somewhere. Anyways. . .the use of medicines also has side effects, right? It will help in some situations but it could interfere in others, right? It's a two-way street. (TGD, SP, FGD)*

> *I was like, I'm not going to take this medicine, it will leave me messed up (. . .) maybe the university is the priority, because if I'm going to use this medicine it might cause side effects and then I'm not going to be able to read, I won't be able to study. (MSM_02, 19 y.o., SSA, interview)*

When asked about PrEP delivery in healthcare facilities, some participants mentioned potential constraints related to providers' judgmental attitudes, discriminatory behavior and anticipated lack of confidentiality based on their previous experiences related to PEP and HIV testing.

> *[referring to his interaction with the doctor when trying to access PEP] I waited for hours at the emergency room when the doctor called my name. His language [referring to biomedical terms] was very difficult to understand. I was trying to explain what had happened to me and he was such a big asshole, talking nonsense. He meant I couldn't have had sex without a condom. And he didn't even know about PEP! Then the pharmacist had to walk him through filling out forms. . . (MSM, BH, FGD)*

## Pharmaceuticalization and trans necropolitics in the relationship between the state and the market: A critical vision about PrEP

A group of the participants' narratives showed that their acceptability of PrEP was shaped by their social positions and life contexts. These differences were expressed in terms of the social inequalities they observed in their personal experiences, family and social environment.

In this sense, the perspective of TGD participants was significantly different from that of MSM. Among the *travestis* and TGW, PrEP could mean just a prophylactic measure that would be insufficient considering the experiences of transphobia, family exclusion and structural racism that they faced given that all of these situations would require direct action by the State to confront them. According to these participants, PrEP and HIV prevention more generally were a secondary worry to many considering that the life expectancy for *travestis* and TGW in Brazil was only 35. A perception that PrEP underscored cisgender and white people's privileged access to prevention also coexisted with this narrative. The quote below expresses the magnitude and complexity of this issue.

> *So, PrEP and PEP are for who? Who has access to them? Who can have the luxury of. . . also. . . anyways. . .(. . .) I think that it was produced for people who fuck around, people with white privilege, that are cisgender, right, that. . .so I think that these problems also have to be solved with the other forms of prevention, and not just put out there, like, this palliative measure and leave the population to keep talking medicine for their entire life and die without any*

*protections, without the State saying anything, without. . .anyway. . .these other groups of society saying anything, right? (. . .) (TGD, SP, FGD)*

Some of the narratives from the focus groups with TGD adolescents indicated distrust towards the pharmaceutical medical apparatus and the efficiency of the State in addressing the health needs of the transgender population. They emphatically criticized isolated measures such as PrEP when compared to the absence of guarantees for basic life needs and rights such as education and health for transgender people.

*I also think that. . .like. . .the AIDS market, right, is part of a genocide process against certain populations, especially the black population, so this [referring to PrEP]. . .anyway. . .in the same way as trafficking, AIDS generates a lot of money for the pharmaceutical companies, for the people that research it, for the institutions that are worried in a certain sense, or not, about treating it. And the distribution in Brazil is free, right. So it's also about how much these medications are able to be inclusive and handle the entire population that is interested in prevention and how much is just about moving money, so [you] also have to think from this point of view, right? What is called necropolitics, this idea that. . .anyway. . .that if the problem exists, it exists for a motive. If there is a population that is dying, in higher numbers due to this problem, prevention is for who? [TGD, SP, FGD]*

Participants, and especially the *travestis* and TGW, also problematized the contexts of access to PrEP and how healthcare facilities delivered it. They emphasized that prevention information arrived disconnected from the life contexts of transgender people and so the benefits that it could bring were never clear. PrEP, like other prevention methods, was seen as being shared through prescriptive messages such as "do this", without addressing broader concerns. They reiterated that for transgender people, messages that addressed issues such as body care and an increase in life expectancy were necessary. As such, they consider that community health workers could offer information about prevention methods other than condoms, in addition to ensuring that this information arrived at the places where these people were (on the streets) and communicated in a way that they understood.

*It's a question of discrimination, the lack of visibility, the lack of a health post, I wouldn't say trained health professionals, but of professionals with the capacity to help us with what we need. The need is very big.' (TGW_02, 16 y.o., BH, interview)*

*I speak pajubá (a transgender dialect), so the least I can do is to bring this language that exists, that is known, and make it so that the information is accessible, right? Putting out a thousand-meter-long words and hoping that the girls will understand it won't help at all and even if she understands, like is she going to take her time to go and want to know more about it, right? It has to be accessible information. . . and to be accessible it has to be part of her reality. (TGD, SP, FGD)*

As such, the perceived effectiveness of PrEP was relativized by social inequality, creating an important counterpoint to the idea of self-efficacy.

## Stigmas related to HIV, adolescence and sexual orientation

In some narratives, especially among MSM, PrEP use was associated with the possibility of being pre-judged as someone who was irresponsible, promiscuous or infected with HIV. Therefore, they referred to fear of discrimination in their family and social circles if anyone

were to find out that they used PrEP. The narratives showed a strong concern about being confused with people who were in HIV treatment in addition to revealing aspects of their sexual life, especially their sexual orientation.

> *My mom doesn't accept it (referring to homosexuality) at all. If she discovered that I take a medication like this one to HIV prevention, because I have sex with other men, geez, I would end up living under a bridge. (MSM, BH, FGD)*

> *At first they (referring to family members) are going to think right away that I'm infected. Because people are very prejudiced. . . it's just that I wouldn't tell them. I could only tell my friends. But you feel right away people backing away. Because they think that you are taking it, to prevent, but that you are infected (. . .) And it's mainly LGBTs. I think that prejudice among LGBT is huge. (MSM_02, 18 y.o., SSA, interview)*

This concern over being judged by others impacted the decision to seek out PrEP. Although some participants referred being interested in using PrEP prior to the interview, none of them actually sought out a clinic. One of the aspects that influenced their decision was the possibility of one of the health professionals contacting their parents due to their being under 18.

> *(Referring to how he knew about PrEP) it was a person that I knew too, that explained it to me, I was interested, it's just that I was afraid because I was. . . until recently I was underage. I was afraid that, who knows, that someone would tell my parents, my uncle, things like that. Then after I turned 18, I sought out the testing center, but it was on strike. (MSM, SP, FGD)*

Participants also shared a perception that, as young people are more subject to prejudice and, therefore, to reproduce it, PrEP would be likely seen as an untrustworthy medication, thus compromising their willingness to PrEP use.

> *It would be a long process to explain [referring to PrEP to young people]. They would have to explain the entire process and, even then, young people wouldn't understand or wouldn't want to understand, due to this prejudice that has been given to them, they simply aren't going to accept taking the medication (. . .). Questions would come up, like 'is it true?' 'Has it been proven?' They don't want it because they'll say it's a lie, so they won't feel motivated because doubts will always come up, it is a prejudice that themselves created or that was given to them, so they aren't willing to take the medication. (MSM_08, 19 y.o., SSA, interview)*

On the other hand, acknowledging young people as being inconsequent about sex was precisely mentioned as why some participants regarded PrEP appropriate for this population.

> *Oh, I would recommend PrEP to young people because it's a time when they're just dum, they do things just because, without thinking of the damage, you know? So re-education and prevention are the best things for young people. (. . .) Transgender girls are careless about taking meds and end up not taking at all. (TR 03 19, SSA, interview)*

Participants also worried about the overlapping of PrEP, sex life and social positions as others evaluated aspects of one's sex life in an unequal way in terms of social, gender and race positions. For example, more negative views could reinforce stigmas and reflect gender (identity and sexual orientation), social class and generation discrimination.

*But it is all the same, you, as a white, cis woman, people are going to say oh cool, she's using it, but it's because she has money to be using that. Now a trans, black woman who lives in the outskirts, if she's using it, of course the stigma associated with it will spread much more easily there. (TGD, BH, FGD)*

As such, AIDS and sexuality stigmas, modulated by social differences (class, ethnicity, race, gender), were important burdens of PrEP that compromised both self-efficacy and adherence to the medication.

## Discussion

The low knowledge and acceptability of PrEP encompasses a normativity characterized by rigid rules for prevention behavior, as well as the social position of adolescent MSM and TGD adolescents. Building from the TFA and in dialogue with the current literature, we discuss our findings based on three main aspects around the prospective acceptability of oral daily PrEP: (i) making sense of new and complex knowledge; (ii) acknowledging that adherence to the 'ideal' method (i.e. condom) is a grey area when it comes to deal with the complexity of actual sexual encounters; and (iii) recognizing social and structural barriers of access to healthcare, in particular discrimination, as key challenges to be overcome to deliver PrEP to most vulnerable adolescents.

### Making sense of new and complex knowledge

A lack of prior knowledge and access to simplified and intelligible information contributed to participants' doubts regarding PrEP effectiveness in terms of its level of protection and what it can 'do' to their body. The most challenging aspects of PrEP were its daily use and feared side effects. Other studies have also found that young people perceived low levels of self-efficacy in managing their routine to take PrEP on a daily basis [27, 28]. Similarly, the anticipation of adverse effects resulting from its use, including its incompatibility with hormone treatment, also demonstrate that a large burden on the individual mediates motivation to use PrEP. Participants' narratives indicate the need for strategies to support young people in organizing their routines to promote PrEP adherence and continuation.

This understanding of PrEP must be contextualized by participants' low or even nonexistent knowledge of the prophylaxis. In this sense, the opportunity costs defined in the TFA model appears to be high at first, given that adolescents need to assimilate a lot of objective information about PrEP (e.g. its prescription, level of protection and interaction with drugs or other medications) that they necessarily relate to their own experiences and perceptions about how taking a medication on a daily basis would be like. As such, participants problematized the possible impact of the continuous use of medication on the body, in the short and long term. In the short term, their concern is represented by the side effects, which would be concealed by adopting strategies to maintain a healthy diet. In the long term, concerns relate to fear of oncoming illness or of limiting their social ascension. The continuous use of PrEP pairs with a chronic disease treatment that would be present throughout life and possibly create dependence. Thus, in MSM's narratives and, in a smaller way, among *travestis* and TGW, there is a future that needs to be considered and protected. In both considerations about present and future, there is an evident concern about opportunity costs evaluated by the identification of possible permanent consequences in their biological and social bodies. These meanings and consequences also deal with the notion of future as part of the value system [29] of these adolescents who identify a threatened future across the notion of dependence. So, the possible addiction caused by continuous use of PrEP co-exists with a fear of physical dependence and

the stigmatization of drug users. Here, the long-term of PrEP use is perceived to lead to a chronic condition, thus highlighting similar aspects described in qualitative studies in which negative beliefs about HIV/AIDS and ARV effectiveness were key barriers to adherence to ARV treatment [30]. On the other hand, it is noteworthy that adolescents' concerns about protecting their future do not include dealing with the consequences of ARV treatment in case they were infected with HIV or having their lives threatened by AIDS. Thus, these young MSM and TGD adolescents seem to be concerned about the future, that include reflecting on the pragmatic consequences and symbolic meanings of their possible use of PrEP. The anticipated stigma and concerns issues about physical consequences played a major role in these concerns. The strength of this role depends on the place of the future as an important notion for them. We noted that the role of near future was stronger among TGD adolescents as most of them emphasized the lack of opportunities and the extremely vulnerable contexts of their life.

## Acknowledging that adherence to the 'ideal' method is a grey area when it comes to deal with the complexity of the actual sexual encounters

Young people predominantly saw PrEP effectiveness as being compared with condom use. The predominant logic of the narratives is that condoms are the most efficient form of protected sex. As such, their occasional or recurrent non-use is signified as a 'mistake' that could be repaired with PrEP. These 'mistakes' are mainly associated with situations that make planning prevention in casual sex between MSM difficult. In the context of prostitution among TGD participants, higher pay for sex without condoms (with possible cocaine use) was frequently mentioned, which has been documented as an advantage of using PrEP among TGW sex workers [31].

The affective attitude towards PrEP exposes a tension between the impossibility of using condoms in all sexual encounters in real life, the association of PrEP as a palliative or an alternative method, and a horizon of greater security, liberty and pleasure offered by the prophylaxis. As such, the narratives express a back-and-forth between a rational and informed reasoning of PrEP efficacy (that reproduces public health's logic) and a moral one based on right and wrong assumptions, such as the perception that 'PrEP isn't good because it only protects against HIV.' The narratives framing PrEP as 'right' or 'effective' highlighted young people's autonomy to seek their "peace of mind" or to feel "secure" in sexual encounters, which can be associated with the concept of ethicality of TFA in the sense that individuals express their position about PrEP by validating or placing into perspective the public health logic.

Hence, PrEP acceptability highlights both visions of hierarchy and complementarity of prevention methods. Considering the historical and social construction of condoms as the 'ideal' method, the effectiveness of PrEP is more strongly associated with intention to use condoms consistently than high adherence to the medication. In this sense, we identify two logics of the PrEP-condom binary that express the perceived effectiveness (TFA) with greater or lesser chances of success based on an affective attitude (TFA) associated with sex.

On the other hand, those that highlighted pleasure as a central dimension of sex referred to PrEP as an option for protected pleasure as it would permit not using condoms. Despite the idealization of condoms, there is a recognition that they may not be used when the aim is to maximize sexual pleasure. However, other prevention strategies and risk management (including those beyond biomedical methods) may occur following one's desires and erotic and cultural experiences [32]. As shown in the narratives presented here, and as noted in the literature [33], PrEP could increase sexual satisfaction and sensations of more security and tranquility given the difficulties surrounding condom use. In this sense, it is not only about a method

being more or less 'complete' or more or less 'adequate'. Methods must be thought of as being part of concrete situations in which people live and have relationships, needs, values, and specific ways of life.

As the majority of participants had no prior knowledge of PrEP, the idealization of condoms as prevention could be, in large part, reflective of them being the only standard they knew. As a consequence, these young people may not necessarily oppose PrEP and idealize condoms, as their prevention mindset in based on one single method and the idea of combination prevention has not become part of their prevention discourse yet. Thus, our findings point towards the processes involved in individuals becoming able to regard something as acceptable, i.e., from getting in contact with information about PrEP to then reflecting on its pertinence and feasibility in daily life.

In terms of moral aspects, it is important to consider how PrEP is also associated with the stigma of promiscuity attributed to gays [34], which relates to the notion of 'risk group' that characterized human rights violations as part of AIDS history. The reproduction of the "risk groups" image seems to impact the acceptability and use of PrEP. As Race [35] discusses, some of the "resistance" in terms of its use are connected to an image of uncomfortableness in relation to risk. The image of "unleashed homosexuality" could provoke doubts and "fear" that with PrEP, what appeared to be an "exception" ends up being the "rule", in the sense of sex "without breaks" among men coming back. Race emphasizes that PrEP acquires the form of a "reluctant object" understood to be the one that confronts a status quo, provokes aversion and even is condemned [35].

Finally, fearing the association of using PrEP with living with HIV demonstrates how HIV is still stigmatized and generates exclusion and a hierarchy among people according to their serostatus, thus impacting PrEP acceptability. In this way, fear of HIV-related stigma could constitute a burden for those who anticipate this condition and have to hide its use, as seen with other uses of ARVs for prevention [36]. The stigma surrounding adolescence may also influence the perceived effectiveness of PrEP as adolescents are discredited in terms of their capacity to choose and use the method with autonomy. The profound social vulnerability and transphobia experienced by TGD people also call into question the effectiveness of PrEP as a method considering its low chances of success in such conditions.

Social vulnerability and the stigmas related to HIV, to adolescence and to transgender identity also refer to ethicality as they form a group of norms and values that need to be problematized for PrEP to be acceptable for young people. Both their skepticism and experiences of stigma highlight how it is important not to lose sight of the social relations that construct certain lives as "lives that don't matter" [37]. Indeed, as the narratives of this article suggest, this aspect of vulnerability affects the access to new prevention services and technologies, in addition to the conditions and ways to negotiate condom use, or even other prevention strategies.

### Recognizing social and structural barriers of access to healthcare, in particular discrimination, as key challenges to be overcome to deliver PrEP to most vulnerable adolescents

The broader context where adolescents imagine PrEP delivery is the dimension most strongly related to the domain of ethicality (TFA). The expressed feelings and attitudes of avoidance, embarrassment and displacement at health services corroborate the perception that healthcare facilities frequently discriminate young people, are transphobic, are not sensitive to contexts underlying sexual relations and to stigmas faced by vulnerable populations, do not use accessible language and, especially in the case of trans people, do not provide comprehensive care.

On the one hand, similar to other studies [38], the prophylaxis is perceived as a collective benefit for the LGBT population, which over and over again is the target of stigma,

discrimination and frequently considered as "responsible" for the HIV epidemic. In this regard, PrEP is affirmative as a right to health and also as freedom from gender- and sexuality-based discrimination. On the other hand, PrEP is perceived with distrust, especially by TGD participants, and as a necro-biopower strategy [39, 40], or rather as a mechanism through which the State defines who (and when) should live and die. In a similar fashion, the meaning of PrEP for young people can also be signified through the lens of medicalization and pharma-ceuticalization and, as consequence, PrEP is seen as a form of control of the transgender body and, by associating certain bodies with certain pharmaceutical remedies, also risks reinforcing a broader tendency of individualizing guilt for social disorders and reducing them to 'a pill for each social ill' [41]. These aspects reinforce the need for coherence in PrEP interventions, as it is perceived differently by people, depending on their gender identity, sexual orientation, social class, and generation.

It is interesting to note that this critical view regarding institutional racism historically per-petrated by the State was shared in a focus group with TGW and *travesti* activists that form part of an LGBT collective that studies trans necropolitics [42] in São Paulo. For them, PrEP represented one more medicalizing element that transforms social questions into medical questions and maintains the white and cisgender population alive through their broader access to the medication. In this regard, they saw the inclusion of PrEP in the Brazilian public health-care system to primarily attend to this population needs as well as the interests of pharmaceuti-cal companies, thereby reinforcing the mechanisms through which material and symbolic resources are concentrated. This critical voice was not unanimous in the groups and inter-views, but it certainly raises important questions for the analysis of discourse and historical processes brought up by PrEP and its acceptability.

## Study limitations

As most participants did not have prior knowledge about PrEP, the impact of dealing with new information also needs to be considered from a methodological perspective, as we did not conduct new interviews and focus groups with the same participants to evaluate if the per-ceived effectiveness and opportunity costs would be similar. As our findings show a diversity of personal, social and structural contexts in which sexual relations and HIV prevention take place, additional research is needed on the acceptability of event-driven and injectable PrEP, as well as acceptability of community-based, simplified and decentralized delivery of PrEP and combination prevention strategies among young people and adolescents.

## Conclusions

Connections between offering PrEP and structural interventions–such as strategies to address poverty, racism, homo and transphobia, and HIV- and adolescence-related stigmas–is decisive to PrEP acceptability, especially among TGD people and other highly vulnerable populations. Despite the biotechnologies and biomedical advances, in Brazil many difficulties and tensions persist in the daily life of those groups, including difficulties in affective-sexual relationships. Certainly, as discussed in this article, social barriers and vulnerabilities, such as stigma and the stereotype of promiscuity, have impacted on PrEP acceptability and it might affect its daily use.

Evidence from PrEP demonstration projects in Brazil has depicted a highly mobilized gay community with access to information and high socioeconomic status that actively demanded PrEP be offered through the public health system [43]. On the other hand, one of the main challenges for expanding the offer of PrEP has been reaching highly vulnerable population groups, such as young people, female sex workers, TGW, and drug users [43]. It is critical that

the offer of PrEP to adolescents and young people be designed to include community mobilization and demand creation strategies. The State must also work to mitigate legal and normative obstacles to accessing health services in a way that they can become socially safe places, especially for underage young people.

## Supporting information

**S1 File. Interview guide for adolescent key informants who participated in the formative research of PrEP 1519 study.** São Paulo, Belo Horizonte, Salvador. Brazil, 2019. (DOCX)

**S2 File. Focus group guide for adolescent key informants who participated in the formative research of PrEP 1519 study.** São Paulo, Belo Horizonte, Salvador. Brazil, 2019. (DOCX)

## Acknowledgments

We are grateful to the adolescents for participating in this study and to Fundação Oswaldo Cruz (Fiocruz) and Fundação de Apoio à Fiocruz (FIOTEC) that provide management support to the study.

## Author Contributions

**Conceptualization:** Eliana Miura Zucchi, Marcia Thereza Couto, Marcelo Castellanos, Alexandre Grangeiro, Inês Dourado, Laio Magno.

**Formal analysis:** Eliana Miura Zucchi, Marcia Thereza Couto, Marcelo Castellanos, Érica Dumont-Pena, Dulce Ferraz, Thiago Félix Pinheiro, Alexandre Grangeiro, Leo Pedrana, Laio Magno.

**Funding acquisition:** Alexandre Grangeiro, Inês Dourado.

**Investigation:** Eliana Miura Zucchi, Marcia Thereza Couto, Marcelo Castellanos, Thiago Félix Pinheiro, Alexandre Grangeiro, Inês Dourado, Laio Magno.

**Methodology:** Eliana Miura Zucchi, Marcia Thereza Couto, Marcelo Castellanos, Érica Dumont-Pena, Dulce Ferraz, Laio Magno.

**Writing – original draft:** Eliana Miura Zucchi, Marcia Thereza Couto, Marcelo Castellanos, Érica Dumont-Pena, Dulce Ferraz, Thiago Félix Pinheiro, Alexandre Grangeiro, Luís Augusto Vasconcelos da Silva, Fernanda Soares de Resende Santos, Laio Magno.

**Writing – review & editing:** Eliana Miura Zucchi, Marcia Thereza Couto, Marcelo Castellanos, Érica Dumont-Pena, Dulce Ferraz, Thiago Félix Pinheiro, Alexandre Grangeiro, Luís Augusto Vasconcelos da Silva, Inês Dourado, Leo Pedrana, Laio Magno.

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
