## [Decision Letter · Decision Letter 0]

14 Dec 2020

PONE-D-20-20709

Acceptability of daily Pre-Exposure Prophylaxis for adolescent men who have sex with men, travestis and transgender women in Brazil: a qualitative study

PLOS ONE

Dear Dr. Zucchi,

Thank you for submitting your manuscript to PLOS ONE. After careful consideration, we feel that it has merit but does not fully meet PLOS ONE’s publication criteria as it currently stands. Therefore, we invite you to submit a revised version of the manuscript that addresses the points raised during the review process.

You need to address the issues of narrative and to improve the flow of the paper.

We look forward to receiving your revised manuscript.

Kind regards,

Andrew R. Dalby, PhD

Academic Editor

PLOS ONE

Journal Requirements:

2.Thank you for stating the following in the Acknowledgments Section of your manuscript:

[We are grateful to the adolescents for participating in this study and to Fundação Oswaldo Cruz (Fiocruz) and Fundação de apoio à Fiocruz (FIOTEC) that support the study.]

 [ID and AG are principal investigators of the PrEP 1519 Study (‘A Demonstration Project of the effectiveness HIV Pre-exposure Prophylaxis (PrEP) amongst adolescent men who have sex with men, transgender women at high risk for HIV infection in the context of Combination Prevention in Brazil’) in the cities of Salvador and São Paulo, respectively, which is funded by Unitaid (grant number 2017-15-FIOTECPrEP).

The Brazilian Ministry of Health, through the Department of Chronic Diseases and Sexually Transmittedy Infections, donates PrEP medications, condoms, and rapid tests.

Unitaid website: https://unitaid.org/#en

Brazilian Ministry of Health website: https://saude.gov.br/

We declare that the funders had no role in study design, data collection and analysis, decision to publish, or preparation of the manuscript.]

3. Please include your tables as part of your main manuscript and remove the individual files. Please note that supplementary tables (should remain/ be uploaded) as separate "supporting information" files.

Reviewers' comments:

Reviewer's Responses to Questions

**Comments to the Author**

1. Is the manuscript technically sound, and do the data support the conclusions?

Reviewer #1: Partly

2. Has the statistical analysis been performed appropriately and rigorously? 

Reviewer #1: Yes

3. Have the authors made all data underlying the findings in their manuscript fully available?

Reviewer #1: Yes

4. Is the manuscript presented in an intelligible fashion and written in standard English?

Reviewer #1: Yes

5. Review Comments to the Author

Reviewer #1: Please see attached comments.

6. PLOS authors have the option to publish the peer review history of their article (what does this mean?). If published, this will include your full peer review and any attached files.

Reviewer #1: No

---

## [Author Response · Author response to Decision Letter 0]

4 Mar 2021

EDITOR’S COMMENT: “You need to address the issues of narrative and to improve the flow of the paper.” 

Response: Thank you for pointing this out. We entirely agree with this comment and, as such, revised the whole text very carefully, especially aiming at more clarity, coherence and concision. Grammar, punctuation, and style were revised as well. In the Methods section, we added subsections (‘Participants’, ‘Materials and Procedures’, ‘Data Analysis’, and ‘Ethical Statement’), and changed the subsection titles in the Discussion in order to improve the flow of the paper. 

REVIEWER'S COMMENTS

Comment 1: I will suggest the authors to review the manuscript and its English grammar as, particularly in the results sections, sometimes is a bit difficult to follow. Sentences can be polished and shorten. I provide some examples and it needs to be edited as a whole.

Response: We agree with this and have incorporated your suggestion throughout the manuscript. As pointed out in the previous response to the Editor’s comment, we have revised the whole text very carefully, especially aiming at more clarity, coherence and concision.

TITLE

Comment 2: Is appropriate and clear, I think it will read better if it has “among adolescent…” rather than “for adolescent”. 

Response: Agree. We replaced the preposition in the title. 

INTRODUCTION

Comment 3: Is well organised and all relevant information is provided. I appreciate that the authors have explained that in Brazil, as in many South-American countries, the word travesti is linked to a political identity and part of the LGBTi community vindicate this identity. 

Response: We appreciate the positive feedback on this.

Comment 4: I would suggest to add some lines on the public policies of Brazil in terms of ARV and PrEP provision. In other countries, not having universal access and gratuity can be considered to be the main barrier, however, this manuscript shows that there are many other issues and misconceptions to work with young people and sexual minority groups.

Response: Thank you for raising this point out. We added Brazil’s public health policy related to HIV and this can be found on page 4, lines 125-129.

Comment 5: In line 63, authors mentioned “The literature describes different socio-structural, personal and programmatic barriers to adolescents’ access to HIV services…”, it seems that authors describe 3 types of barriers, but according to my understanding and the description provided, programatic barriers should be included in structural barriers… in line 70, interpersonal barriers, that were not mentioned in that introductory sentences, are described… perhaps this can be polished and talk about personal, social and structural barriers… or another categorisation but provide the description using same category.

Response: We agree with this comment. We rewrote the entire paragraph in order to ensure more clarity, specially emphasizing programmatic barriers as structural barriers. The modified paragraph can be found on page 3, lines 64-74. 

Comment 6: Line 84-85 review the sentence “high acceptability of PrEP has been observed in many investigations, and facilitators to it include free access to PrEP”… “facilitators to it include” may be replaced by “the main facilitators were” or another wording.

Response: Agree. We replaced it by ‘main facilitators were’ (page 4, line 123).

Comment 7: Line 89-90 authors mentioned “The meanings that the adolescents attribute to processes relevant to the acceptability of PrEP include…”, however, the description includes a series of aspects, such as sexual orientation or autonomy, that are not meanings of… probably this sentences can be rewrite and simplify… “Relevant aspects/issues/ topics…. For prep acceptability among adolescents are…. “

Response: We agree with this suggestion, which was incorporated to the text on page 4, lines 131-133.

Comment 8: In line 95, authors mentioned “life contexts and sexual relationships”… as it is currently written, it seems as a separate set of variables… but I understand that authors intended to explore the role of life contexts and sexual relationships on Prep acceptability. I suggest to rephrase this sentence

Response: We rephrased this sentence to ‘as well as how their sexual encounters intersect with their life contexts,’, which can be found on page 4, lines 137-138.

METHODS

Comment 9: Overall is very complete but it is difficult to follow… I would only suggest to add some subtitles to organise the information: participants, materials/instruments, procedure, data analyses, ethical considerations/statement…or the like.

Response: We agree with this comment and, therefore, we added four subsections (‘Participants’, ‘Materials and Procedures’, ‘Data Analysis’, and ‘Ethical Statement’) in the Methods section (pages 4-9).

Comment 10: Authors mention 1) mapping, 2) interviews and 3) focus-groups…. Can you describe better how you organised the mapping?

Response: This explanation is provided on page 5, lines 163-169.

Comment 11: It is not quite clear how participants were recruited . The section starts with “after mapping out spaces of social interaction though participant observation”, which is not vey clear,… and finished with Participants were recruited in public and private high schools, and adolescent LGBT venues. Can you clarify the link or sequence between “mapping spaces”, “participants observation”, “schools and venues”,…

Response: Participant observation was part of the mapping of adolescent venues, which was followed by interviews and focus group discussions. We added a more detailed description that can be found on pages 5-6, lines 163-181, in the sentences below:

As a first step, in each city, we mapped out adolescent venues with a high concentration of MSM, travestis and TGW, both in face-to-face settings (e.g., beaches, parks, bars, parties, sports and leisure facilities, schools, health and human rights services) and in online spaces (e.g., social media, internet, and dating apps). Participant observation was conducted in order to understand the youth venues dynamics and to identify potential adolescent key-informants. As a second step, between August 2018 and January 2019, we selected – mainly at bars, parties, schools and dating apps – key-informants to participate in semi-structured interviews and in FGD in the three aforementioned cities.

Comment 12: In Table 1, the word “transvestite” rather than travesti is written, It would be better to change it for consistency.

Response: Thank you for your observation. We corrected it for travesti on Table 1.

Comment 13: Interviews and focus groups used the same methodology?  the interview guide was the same for both interviews and goops?

Response: Interviews and focus group discussions had a common set of questions about PrEP that provided data for the acceptability analysis. Interviews also focused on more sensitive and private issues of adolescents’ life contexts that were related to PrEP acceptability and HIV prevention as a whole. We included the guide for focus group discussion (Table 2) as part of this revised version of the manuscript. The guide is first mentioned on page 6, lines 209-210, and can be found on pages 43- 47.

RESULTS

Comment 14: The section is quite long and it can be easily shorten by removing narratives that point to the same explanation. There also are repetitive explanations, for example with transactional sex, that can be presented in one paragraph and provide one or two narratives as examples.

Response: We agree with this comment and sought to restructure this section focusing on concision. 

Comment 15: In the description of participants, both in text and in Table 2, bisexual, asexual, etc.. have double “s”. Ej: panssexual, should be pansexual, bissexual, should be bisexual...

Response: Thank you for this observation. We corrected the spelling of this paragraph on page 10, lines 308-309. 

Comment 16: Lines 225-228, the explanation is clear but the three examples are quite similar and non of them refer to doubts of drug interaction, for instance. I suggest authors to change some of the reiterative narratives and replace for a diferente example. 

Response: We added the following quotes (page 11, lines 343-347) to illustrate this data, as below:

There’s counterindication so, for example, when you take PrEP, you can’t use drugs. (MSM, SSA, FGD)

I’m starting to use hormones, I have to take three pills a day. So I was wondering if PrEP cuts the effect of hormones, what its side effects are. (TGW, SSA, FGD)

Comment 17: Lines 251-252, talk about prejudices and misconceptions related to PreP use that act as barriers to its accessibility, however, the example provided, is a bit long and reiterative and not very representative of the interpretation. This can be polished.

Response: We agree with this comment. We rephrased and then moved this paragraph and quote to the subsection “Stigmas related to HIV, adolescence and sexual orientation’ as they refer to the impact of discredit of adolescents when they get in touch with information about PrEP.

Comment 18: Line 263 “credited PrEP with additional protection against STIs” I will clarify that this is a misconception.

Response: We added this information on page 12, lines 374-375.

Comment 19: Line 275… review the wording of the following sentence “ especially with casual sex as this raised more concern about STIs” …. “be it because”

Response: We rephrased this these sentences, which can be found on page 12, lines 387-390, as below:

“especially in the context of casual sex, which was perceived as posing higher risk for STIs. The ‘ideal’ use of PrEP appeared together with condom use, either because condoms were seen as being indispensable by themselves or because participants considered the possibility of getting infected with other STIs.”

DISCUSSION

Comment 20: This section is very interesting, particularly in the discussion of ideas, meditating process and the like. But it seems to be written by a different person, different English, writing style… and also, some of the thinking provided, are not well presented in the results section. Thus, I will suggest to review the coherence between both sections.

Response: We appreciate and agree with this comment. As mentioned in some previous responses, the Discussion section was also entirely revised aiming at cohesion, coherence and consistency of the ideas presented. The first paragraph was modified in order to highlight the general conclusion and to present three central issues around PrEP acceptability that were the basis for the three subsections of the Discussion (page 24, lines 714-722). As well as rephrasing the text substantially, we hope that restructuring the subsections will help provide more clarity and a better flow of the ideas. 

Comment 21: I found the first paragraph not at all supported by the testimonies provided but for the feeling and interpretations of researchaers… “feeling and anguish” …“distrust of sexual healthcare providers”. These ideas seems not to be presented in the results section, or, if they are, it seems to me that they were not the main barriers, at least, not presented in that manner. You can revise the wording. 

Response: Thank you for this comment. In fact, the results did not support such interpretation adequately. Therefore, we included one paragraph and quote in the subsection “Barriers to PrEP use”. The key idea is to highlight that, when participants were asked to think of what PrEP delivery would and/or should be like, many of them immediately brought up unpleasant or discriminatory experiences related to HIV prevention in healthcare facilities. This can be found on page 18, lines 563-573, and as below.

When asked about PrEP delivery in healthcare facilities, some participants mentioned potential constraints related to providers’ judgmental attitudes, discriminatory behavior and anticipated lack of confidentiality based on their previous experiences related to PEP and HIV testing. 

[referring to his interaction with the doctor when trying to access PEP] I waited for hours at the emergency room when the doctor called my name. His language [referring to biomedical terms] was very difficult to understand. I was trying to explain what had happened to me and he was such a big asshole, talking nonsense. He meant I couldn’t have had sex without a condom. And he didn’t even know about PEP! Then the pharmacist had to walk him through filling out forms… (MSM, BH, FGD)

Comment 22: Line 546-47 mentioned “Other studies have also found that young people perceived low levels of self-efficacy in terms of managing their routine to take PrEP on a daily basis” the references for that should be provided precisely in the discussion.

Response: We added two references (Serota el atl, 2020; Bourne et al, 2017) to this paragraph that can be found on page 24, line 768. 

Comment 23: Line 559-561 can be moved to a limitation section

Response: We agree with this suggestion and added a ‘Study Limitations’ subsection on page 31, lines 1169-1179.

Comment 24: The paragraph on “continuous use of medication”, not only is very long, but it is not well articulated with the rest of the discussion neither with the results section… it also seems to be written with a different English. Thus, I suggest to shorten it and probably make some of this interpretations in the results section.

Response: As mentioned in some previous responses, we hope that restructuring and rephrasing this section substantially helped provide more clarity and a better flow of ideas. In relation to how adolescents perceive the continuous use of medication, we sought to highlight short- and long-term consequences associated to PrEP use over time and how these are connected to adolescents’ value system in relation to their perspective of future. This can be found on pages 25-26, lines 773-853.

Comment 25: Similar impression with the next sections of the discussion. 

Response: As mentioned previously, we hope that restructuring and rephrasing helped address this issue in this section. 

Comment 26: A section with limitations and suggestions for future research should be added (short and simple, no more than one paragraph).

Response: We agree with this suggestion and added a ‘Study Limitations’ subsection on page 31, lines 1169-1179. Based on our findings, key suggestions for future research included acceptability investigations on event-driven and injectable PrEP among adolescents, as well as on community-based and decentralized PrEP delivery.

---

## [Editor Report · Decision Letter 1]

16 Mar 2021

Acceptability of daily Pre-Exposure Prophylaxis among adolescent men who have sex with men, travestis and transgender women in Brazil: a qualitative study

PONE-D-20-20709R1

Dear Dr. Zucchi,

We’re pleased to inform you that your manuscript has been judged scientifically suitable for publication and will be formally accepted for publication once it meets all outstanding technical requirements.

Kind regards,

Andrew R. Dalby, PhD

Academic Editor

PLOS ONE
---

## [Editor Report · Acceptance letter]

16 Apr 2021

PONE-D-20-20709R1 

Acceptability of daily Pre-Exposure Prophylaxis among adolescent men who have sex with men, *travestis* and transgender women in Brazil: a qualitative study 

Dear Dr. Zucchi:

I'm pleased to inform you that your manuscript has been deemed suitable for publication in PLOS ONE. Congratulations! Your manuscript is now with our production department. 

Kind regards, 

on behalf of

Dr. Andrew R. Dalby 

Academic Editor

PLOS ONE